# Sobrerol Improves Memory Impairment in the Scopolamine-Induced Amnesia Mouse Model

**DOI:** 10.3390/ijms26104613

**Published:** 2025-05-12

**Authors:** AbuZar Ansari, Geon-Seok Park, Soo-Jeong Park, A-Ra Goh, Kang-Hoon Je

**Affiliations:** NeurolMed, Room 302, 91, Changryongdae-ro 256 beon-gil, Yeongtong-gu, Suwon-si 16229, Republic of Korea; abu.kim.0313@gmail.com (A.A.); geonseok93@gmail.com (G.-S.P.); imation99@naver.com (S.-J.P.); argoh4027@gmail.com (A.-R.G.)

**Keywords:** amnesia, memory, scopolamine, acetylcholine, amyloid beta, phospho-tau

## Abstract

Memory impairment is a defining characteristic of Alzheimer’s disease (AD), with amnesia often appearing as its earliest symptom. Given the multifactorial nature of AD pathogenesis, this study investigates the multi-target therapeutic potential of sobrerol (coded as NRM-331) in a scopolamine-induced amnesia mouse model, focusing specifically on its effects in ameliorating memory deficits and enhancing neuronal plasticity. Sixty male C57BL/6NCrljOri mice were divided into six groups (10 mice/group): vehicle control (CTL, saline), scopolamine (SPA, 10 mg/kg/day), Aricept (APT, 2 mg/kg/day), and three treatment groups receiving NRM-331 at doses of 40, 80, and 100 mg/kg/day. Several behavioral tests were conducted, including the Y-maze test, passive avoidance test, and Morris water maze test. Additionally, biochemical assays were performed in serum (to measure Aß 1-40 and Aß 1-42) and in the brain (to assess ACh and AChE levels), along with histopathological examination of the brain using Nissl staining and p-tau IHC. No significant change was observed in the Y-maze test or the acquisition trial of the passive avoidance test. However, improvements were noted in the retention trial of the passive avoidance test and the Morris water maze test (including escape latency, swim distance, and number of platform crossed) for the NRM-331 groups compared to the SPA group. Serum levels of Aß 1-40 and Aß 1-42 decreased in the NRM-331 groups compared to the SPA group. In the brain, levels of ACh significantly increased, while AChE levels significantly decreased compared to the SPA group. The number of neuronal cells improved in the CA1, CA3, and DG regions of the hippocampus, as indicated by Nissl staining. A significant reduction in p-tau accumulation was also observed in the NRM-331 groups. In conclusion, NRM-331 demonstrated an anti-amnesic effect by enhancing hippocampal cholinergic signaling, alongside exhibiting anti-tau and anti-Aβ synthesis properties. These therapeutic effects suggest that NRM-331 significantly mitigates memory impairment induced by SPA through a neuroprotective mechanism.

## 1. Introduction

Amnesia refers to the loss of short-term memory, particularly the inability to retain facts, information, and experiences, which is a primary symptom of dementia [1]. This memory loss is often the result of neuronal damage or dysfunction of the hippocampus, the brain region responsible for memory formation and retrieval [2]. The causes of amnesia include traumatic brain injury, stroke, inflammatory reactions from infections, oxygen deficiency, excessive alcohol consumption (such as in Korsakoff syndrome), brain tumors, use of sedatives like benzodiazepines in epilepsy, and neurodegenerative diseases like dementia [3]. Alzheimer’s disease (AD), the most common form of dementia, is characterized by the progressive loss of memory, learning, and other cognitive impairments due to the ongoing degeneration of neurons and synaptic dysfunction [4]. Amnesia is typically a temporary and reversible condition, while dementia is progressive and irreversible. Treatment options for amnesia can include non-specific muscarinic anticholinergic drugs, which may aid in recalling events that occurred before brain damage; however, individuals often struggle to remember events after the injury [5]. In contrast, neurodegenerative amnesia, as seen in AD, is characterized by the inability to recall past events or previously familiar information.

The fundamental processes for learning and memory formation, which rely on neural communication, are crucial for encoding, consolidating, and retrieving information. These processes depend on neuroplasticity and the activity of neurotransmitters [6]. Memory loss is a key symptom of AD, as is the accumulation of amyloid-beta (Aβ) plaques and tau tangles, central molecules involved in the progressive damage to neurons [7]. Aβ protein plaques accumulate between neurons and disrupt neuronal communication, while hyperphosphorylated tau protein tangles accumulate inside neurons, which impairs the intracellular transport of neurotransmitters. The accumulation of amyloid plaques and tau tangles exacerbates neuronal inflammation and damage, ultimately leading to neurodegeneration. This neurodegeneration contributes to cognitive decline and memory loss, which are hallmark features of AD-related dementia and amnesia [8]. Moreover, memory impairment is linked to dysfunction in the cholinergic system, which includes cholinergic neurons, neurotransmitters, and their receptors [9]. According to the cholinergic hypothesis of memory dysfunction, decreased integrity of cholinergic neurotransmission, particularly involving acetylcholine (ACh) and acetylcholinesterase (AChE), affects synaptic plasticity, attention, memory, and learning [10]. Disruption of synaptic plasticity is commonly observed in both amnesia and AD; however, it may be reversible through the action of chlorogenic acid, which exerts its effects via anti-AChE and anti-oxidative activities [11].

Alkaloids are pharmacologically characterized as causing sensory loss, which makes it difficult to recall memories. Scopolamine (SPA), a tropane alkaloid, is a potent psychoactive drug that affects cholinergic signaling, as actively investigated in the 1950s [12]. SPA induces both short-term and long-term memory loss by blocking cholinergic transmission through its antagonist action on muscarinic receptors in the brain, thereby impairing learning and memory [13]. Recent studies suggest that natural compounds like *Callicarpa dichotoma* (Lour.) K Koch, couarin, and ginsenosides (Rg5 and Rh3) have a neuroprotective effect against SPA-induced memory deficits [14,15,16]. Additionally, calcium homeostasis is dysregulated in AD, but this can be reversed by *Bacopa Monnier* (L) Wettst through the CREB pathway in SPA-induced amnesia [17].

There are currently no specific medications that can help alleviate symptoms or slow the disease’s progression. These medications may have side effects, including nausea, diarrhea, insomnia, muscle cramps, and dizziness. Aricept (Donepezil), a commonly prescribed medication, works by inhibiting ACh breakdown, thereby enhancing its levels in the brain. This highlights the importance of cholinergic pathways in memory recovery and underscores the need to investigate underlying neurobiological markers, such as amyloid-β and p-tau; both are implicated in the neurodegenerative processes of AD. Sobrerol (C10H18O2) is a synthetic mucolytic agent derived from terpenes, traditionally investigated as an oral mucolytic agent for lung and bronchitis complications [18,19,20]. Sobrerol consists of anti-inflammatory and antioxidant effects, a mechanism of breaking disulfide bonds in mucoproteins, and increased mucociliary transport [21,22]. In addition, it has recently emerged as a potential therapeutic candidate for multiple sclerosis, as demonstrated in contemporary studies [23]. Building upon these findings, we hypothesized that sobrerol (coded as NRM-331) may exert additional previously unreported pharmacological effects. To explore this possibility, the present study aims to rigorously evaluate its potential impact on cognitive dysfunction, thereby expanding the understanding of its therapeutic scope. In this study, we aimed to investigate these factors and their interactions to better understand the SPA-induced amnesia mouse model and to evaluate the potential effects of NRM-331 in mitigating memory impairments and neuronal dysfunction. To achieve this, we conducted several behavioral tests (Y-maze test, passive avoidance test, and Morris water maze test), biochemical analysis in serum (Aß 1-40 and Aß 1-42), as well as in the brain (ACh and AChE) and brain histopathological analysis (Nissl staining and tau-IHC).

## 2. Result

### 2.1. Effect of NRM-331 Treatment on Animals During the Study

Six experimental groups (*n* = 10/group) were examined as follows: saline vehicle control (CTL); negative control with 10 mg/kg/day scopolamine (SPA); positive control with 2 mg/kg/day Aricept (ACT) and three groups treated with NRM-331 at doses of 40 mg/kg/day (N40), 80 mg/kg/day (N80), and 100 mg/kg/day (N100). The pathogen test report of the animal environment indicated that there were no pathogenic factors that could affect the experiment. Body weights were measured on days 1, 8, 15, 22, and 29. Throughout the study period, no significant change in body weight was observed in any of the NRM-331 administration groups (N40, N80, and N100) or the ACT group compared to the CTL group (Figure 1A). However, a significant weight loss was observed in the SPA group on day 29 compared to the CTL group (*p* < 0.05, Figure 1B). Additionally, throughout the study period, no abnormal symptoms or deaths were recorded in any of the NRM-331 administration groups (N40, N80, and N100), indicating no adverse effects from the administration of NRM-331.

### 2.2. Effect of NRM-331 in Behavioral Tests

The impact of NRM-331 on short-term memory was evaluated using the Y-maze test for behavior change on day 7. In the SPA group, a significant decrease in altered behavior was observed compared to the CTL group (*p* < 0.05). However, no significant changes were noted in the NRM-331 administration groups (N40, N80, and N100). Interestingly, the ACT group showed increased and altered behavior (31.77%) compared to the SPA group (Figure 2A). These findings suggest that NRM-331 administration did not significantly affect short-term memory performance in the Y-maze test compared to SPA administration. 

In addition to the Y-maze observations, we assessed short-term learning and memory through the passive avoidance test on days 14 and 15, focusing on the acquisition and retention of trials. On day 14, the SPA group exhibited a significant increase in avoidance time (57.41%) compared to the CTL group. Conversely, no significant changes were observed in the NRM-331 administration groups (N40, N80, and N100), including the ACT group, compared to the SPL group (Figure 2B). On day 15, during the retention trial, a significant reduction in avoidance time was noted in the SPL group compared to the CTL group (*p* < 0.01). The N40 administration group did not show significant differences in avoidance time; however, the N80 administration group demonstrated a significant increase (26.97%), and the N100 administration group approximately showed a 60% increase in a dose-dependent manner when compared to the SPL group (*p* < 0.05 or *p* < 0.01). Additionally, the ACT group exhibited a significant increase in avoidance time compared to the SPL group (*p* < 0.01, Figure 2C). These results indicate that learning was successfully encoded in memory and could be recalled after a delay.

To assess long-term spatial learning and memory, the Morris water maze test was conducted on days 24, 25, 26, and 27. A significant increase in travel time to locate the platform was observed in the SPA group compared to the CTL group (*p* < 0.01) on all test days. In contrast, the NRM-331 administration groups (N40, N80, and N100), as well as the ACT group, showed a significant reduction in travel time on all test days (*p* < 0.01; Figure 2D). A similar trend was noted for travel distance, with the SPA group displaying a significant increase in distance traveled compared to the CTL group (*p* < 0.01). The N40, N80, and N100 administration groups showed a statistically significant decrease in travel distance compared to the SPA group, with the N100 group demonstrating a 15.07% decrease compared to the ACT group (Figure 2E). On day 28, the number of crossings over the platform location after its removal was assessed. The SPA group showed a significant decrease in the number of crossings compared to the CTL group (*p* < 0.01). Conversely, significant increases in the number of crossings were observed in the N40, N80, and N100 administration groups, including the ACT group, when compared to the SPA group (*p* < 0.01; Figure 2F). The representative track maps from the Morris water maze test on day 27 and the cross-number probe trial track map on day 28 illustrate that memory impairment appears complex in the SPA group, while the NRM-331 administration improved this impairment in a dose-dependent manner, as observed in the ACT group (Appendix A).

### 2.3. Effect of NRM-331 on Biochemical Analysis

Amyloid-β peptides 1-40 and 1-42 were measured in serum using an ELISA kit. The results showed that amyloid-β 1-40 and 1-42 levels were significantly increased in the SPA group compared to the CTL group (*p* < 0.01). In the N40, N80, and N100 administration groups, significant reductions in either amyloid-β 1-40 or amyloid-β 1-42 were observed compared to the SPA group. Additionally, in the ACT group, there was a significant decrease in both amyloid-β 1-40 and 1-42 levels, compared to the SPA group (*p* < 0.05; Figure 3A,B). Acetylcholine (ACh) and acetylcholinesterase (AChE) levels were also measured in brain homogenate using ELISA. The results indicated a significant decrease in ACh levels and a significant increase in AChE levels in the SPA group compared to the CTL group (*p* < 0.01). In the N40, N80, and N100 administration groups, including the ACT group, ACh levels showed a significant increase in a dose-dependent manner compared to the SPA group (*p* < 0.01; Figure 3C). Conversely, AChE levels in the N40, N80, and N100 administration groups, including the ACT group, demonstrated a significant decrease compared to the SPA group, which was also in a dose-dependent manner (*p* < 0.01; Figure 3D).

### 2.4. Effect of NRM-331 on Neuronal Plasticity

The histopathological analysis of neuron counts in the CA1, CA3, and DG regions of the hippocampus, using Nissl stain, showed no statistically significant difference in neuron numbers between the SPA group and the CTL group. However, a decrease in neuron counts was observed in the SPA group, with reductions of 16.69% in CA1, 15.34% in CA3, and 12.60% in DG. Furthermore, while there were no statistically significant changes in the neuron counts in the N40, N80, and N100 administration groups compared to the SPA group, increases in neuron counts were noted in the CA1 region for the N80 and N100 administration groups, showing increases of 11.95% and 10.44%, respectively. In the same groups, the DG region also showed increased neuron counts of 12.75%, 11.20%, and 11.42% compared to the ACT group (see Figure 4A,B).

Additionally, the quantitative analysis of p-tau accumulation in the CA1, CA3, and DG regions of the hippocampus was conducted through immunohistochemical staining, using the SPA group as a reference. A statistically significant increase in p-tau levels was observed in the SPA group, with increases of approximately 12.29% in both the CA1 and DG regions (*p* < 0.01). However, the increase in the CA3 region was not statistically significant compared to the CTL group. In both the N40 and N80 administration groups, a significant reduction in p-tau levels was noted in the CA3 region compared to the SPA group (*p* < 0.05 or *p* < 0.01). In the N100 administration group, a significant decrease in p-tau of around 11.16% was observed in the CA3 and DG regions (*p* < 0.01), although the decrease in the CA1 region was not statistically significant compared to the SPA group. Furthermore, in the ACT group, significant reductions in p-tau levels were observed across all three sites (CA1, CA3, and DG) compared to the SPA group (*p* < 0.01; see Figure 4C,D).

## 3. Discussion

Memory loss is a hallmark symptom of AD, a multifactorial disorder characterized by a progressive decline of ACh levels, the accumulation of Aβ plaques between neurons, and the formation of tau tangles within neurons; these factors contribute to neurodegeneration and cognitive decline [9,24,25]. Due to the complex and multifunctional nature of AD, a multi-targeted therapeutic approach is preferred over a single-target drug to effectively address the intricate pathophysiological mechanisms underlying the disease. Current therapeutic approaches in the initial stages of AD often focus on enhancing cholinergic function in the brain, such as using Aricept (Donepezil), which inhibits the activity of AChE, thereby increasing ACh levels. However, the underlying mechanisms of these treatments are not yet fully understood, and they can sometimes have adverse effects [26]. Recently, there has been growing interest in multi-target ligands and natural therapeutic agents as complementary and alternative treatments for neurodegenerative disorders [27]. In the present study, NRM-331 (sobrerol) approached the multifunctional properties of NRM-331 to evaluate its neuroprotective effect in a SPA-induced amnesia mouse model. Our results demonstrate that NRM-331 improves memory and cognitive impairment in this model. The improvements were validated by a reduction in serum Aβ concentrations, an increase in ACh levels in the brain, and a decrease in p-tau accumulation in the hippocampus, all of which reflect enhanced neuronal plasticity due to NRM-331.

The cholinergic antagonist properties of SPA provide insight into the mechanisms underlying memory impairment associated with neurodegenerative disorders and potential avenues for treating amnesia [28]. Previous studies have shown that SPA administration induces cognitive deficits in experimental models of learning and memory impairment [29,30]. To evaluate spatial learning and memory impairment, we conducted the Y-maze test, passive avoidance test, and Morris water maze test during NRM-331 treatment in SPA-induced amnestic mice. The Y-maze test, commonly used to study memory impairments and neurodegenerative diseases, evaluates short-term memory by observing rodents’ instinctive behavior of exploring new environments [31]. Our findings from the Y-maze test indicate that administration of NRM-331 effectively inhibits the deterioration of short-term spatial memory compared to the SPA group. The passive avoidance test assesses short-term learning and memory by creating a conflict between fear and a natural preference for darkness, often induced by electric shock [32]. The passive avoidance tests are often employed to study memory retention, such as in neurodegenerative diseases [33]. In this test, we observed no significant difference in avoidance time at lower concentrations of NRM-331, while at higher concentrations, there was an increased avoidance time compared to the SPA group. This suggests that higher doses of the NRM-331 may enhance memory retention or reduce memory impairment, indicating potential cognitive-enhancing effects. The Morris water maze test assesses the acquisition of spatial memory, making it a valuable tool for studying hippocampal-dependent memory [34]. Over the trial time, the latency to find the platform reflects the animal’s ability to learn and remember the location of the hidden platform, evaluating long-term spatial learning abilities. The results indicated a dose-dependent enhancement in cognitive function, as higher concentrations of NRM-331 (40, 80, and 100 mg/kg/day) significantly reduced travel time and distance, suggesting improved spatial memory and learning abilities compared to the SPA group. Overall, the findings from the Y-maze, passive avoidance, and Morris water maze tests align with previous studies that reported improvements in learning and memory following scopolamine-induced impairments [35,36]. Overall, the administration of NRM-331 significantly ameliorated learning and memory deficits induced by SPA.

Amnesia, often resulting from traumatic brain injury, is closely associated with the Aβ and tau protein aggregation as seen in AD [37]. The complex relationship between neurobiological factors and cognitive functioning is highlighted by the critical roles of Aβ, which serves as a key marker in the pathology of AD. A recent study indicated that the blood Aβ 1-42/1-40 ratio can serve as a predictive marker for the progression from mild cognitive impairment, leading to dementia [38]. Change in Aβ 1-40 and Aβ 1-42 levels can result in a decline in communication abilities and overall cognitive health, manifesting as amnesia and diminished learning capacity. In our study, we observed a decrease in serum Aβ 1-40 and Aβ 1-42 levels in the NRM-331 administration groups compared to the SPA group, indicating that NRM-331 has an effect in reducing these Aβ levels. Learning and memory processes are intricate and involve numerous neurotransmitters and receptors, with cholinergic signaling being one of them. Cholinergic deficits are a significant mechanism in AD, contributing to the decline in cognitive functions, including memory loss [39]. Understanding the interactions between these biomarkers and cholinergic treatment strategies is crucial for developing effective therapies aimed at alleviating the cognitive deficits associated with SPA-induced amnesia [11]. To investigate the role of NRM-331 in neurotransmitter signaling, we examined ACh and AChE levels of cholinergic signaling by ELISA kits. Maintaining proper levels of ACh is essential for normal memory function; however, excessive activity of AChE can disrupt ACh availability in cholinergic synapses located in the hippocampus. In the context of SPA-induced amnesia, higher AChE activity is observed, which negatively impacts memory. Reducing or eliminating this excessive AChE activity may help restore functionality within the cholinergic neuronal system. Our results indicated increased ACh levels in the NRM-331 administration groups compared to the SPA group, along with a significant decrease in AChE levels in the same groups. Previous studies have shown that lower ACh levels can delay object recognition [40]. As mentioned, sobrerol consists of antioxidant properties, and oxidative stress and inflammation are key mechanisms involved in cognitive decline. In the current study, our primary objective was to investigate the effects of sobrerol on cholinergic markers (ACh, AChE), as well as neuropathological markers such as p-tau and Aβ, in relation to behavioral performance. Access to oxidative stress or inflammatory markers (e.g., ROS, IL-6, TNF-α, BDNF, CREB) would have provided a more comprehensive understanding of the mechanisms underlying sobrerol’s cognitive protective effects. We did not assess oxidative stress or inflammatory markers, which is the limitation of this study.

The hippocampus is crucial for memory formation, and memory loss is a principal symptom of both AD and amnesia [41]. Damage to the hippocampus, particularly in the broader medial temporal lobe, leads to deficits in episodic memory, delayed recall, and recollective experience, associated with amnesia [42]. The impact of Aβ and p-tau protein on memory disorders such as AD cannot be overstated, as they contribute to the neurodegenerative processes underlying amnesia. Significant neurodegeneration has been linked to a decline in communication capabilities and overall cognitive health, leading to amnesia and diminished learning capacity. The hippocampus is recognized as central to memory formation and abnormal synaptic circuits and activity. The major input to the hippocampus comes from the CA1 to the subiculum through two main pathways: direct and indirect. In the indirect pathway, information reaches the CA1 region via the trisynaptic circuit. Axons project to the granule cells of the dentate gyrus (DG, first synapse), then continue through the mossy fibers to CA3 (second synapse), concluding the circuit with axons from CA1 (third synapse). Axons from CA1 project back to the entorhinal cortex (EC), completing the circuit. Any dysregulation in synaptic communication due to neurotransmitters (such as ACh) or neuroplasticity resulting from neurodegeneration (like Aβ or p-tau plaque accumulation) can impact the integrity of this trisynaptic circuit. In our study, following the administration of NRM-331, histopathological examination using Nissl stain revealed an increase in the number of neurons in the CA1, CA3, and DG sites of the hippocampus compared to the SPA group. Furthermore, immunohistochemical analysis for p-tau accumulation demonstrated a significant reduction in the NRM-331 administration group relative to the SPA group.

Conclusion: In conclusion, our findings provide evidence that NRM-331 exhibits an anti-amnesic effect, potentially attributable to its ability to enhance cholinergic signaling in the hippocampus, as well as its anti-tau and anti-Aβ synthesis activities. These properties suggest that NRM-331 could serve as a promising therapeutic agent for patients with neurodegenerative disorders, such as AD, by addressing key factors contributing to memory impairment and neuronal degeneration. This could pave the way for the development of multifunctional treatments featuring NRM-331 for AD and other related cognitive disorders.

## 4. Materials and Methods

### 4.1. About NRM-331

Sobrerol (trans-p-Menth-6-ene-2,8-diol) is an aliphatic homomonocyclic compound (coded as NRM-331). One of the key mechanisms of action of Sobrerol is its antioxidant activity. Antioxidant activity is important for evaluating the capacity of compounds to reduce reactive oxygen species, thereby inhibiting oxidation and the production of free radicals. In this study, we assessed the neuroprotective effect of NRM-331 using the Scopolamine-induced amnesia mouse model.

### 4.2. Animal Study Design

Male C57BL/6NCrljOri mice (*n* = 91, eight weeks old) were purchased from Orient Bio Co., Ltd. (Seongnam-si, Republic of Korea). Upon arrival, the body weights of the animals were measured, ranging from 19.86 to 23.81 g. Five animals were housed per polycarbonate rearing cage (W 200 × L 260 × H 130 mm), with bedding consisting of sterilized wooden flakes (Saron Bio, Uiwang-si, Republic of Korea). Animals were acclimatized for seven days in a specific pathogen-free (SPF) environment, with free access to food and water. A solid rodent diet (Teklad global diets, 18% protein rodent diet, 2918C, ENVIGO, Indianapolis, IN, USA) was sourced from Coretech (Pyeongtaek-si, Republic of Korea), and ultraviolet sterilized water was provided in a polycarbonate drinking bottle with a microfilter. Water quality was tested by the Gyeonggi-do Institute of Health and Environment (Suwon-si, Republic of Korea). The animal room conditions were controlled at a temperature of 22 ± 3 °C, with relative humidity maintained at 55 ± 15% and a ventilation frequency of 10–20 times per hour. A 12-h light (08:00 a.m. to 08:00 p.m.) and 12-h dark (08:00 p.m. to 08:00 a.m.) cycle was implemented, with illuminance measured at 150–300 Lux. Throughout the study, temperature and humidity were recorded hourly using a computer system, while ventilation frequency and light levels were measured regularly. Bedding and water were checked daily, with bedding changed once a week. The animal room was routinely monitored for pathogen testing. Following the acclimatization period, a cued test was conducted to select healthy animals and exclude those deemed unhealthy, which is essential for reliable study outcomes. The pool was filled with water, with the platform submerged just below the surface. When placed in the pool, animals had 60 s to locate the platform; those that failed to find it were excluded from the experiment after two attempts. Following the cued test, 60 healthy animals were selected, weighed, marked, and randomly assigned to six experimental groups (*n* = 10 mice/group) to ensure uniform body weight distribution (ranging from 21.88 to 24.94 g). After grouping, similarly, five animals were housed per cage and cages were labeled with unique identifiers and color codes: G1 (M 10 1–10—10 Saline), G2 (M 10 11–20—10 Scopolamine), G3 (M 10 21–30—10 Aricept), G4 (M 10 31–40—10 NRM-331 at 40 mg/kg/day), G5 (M 10 41–50—10 NRM-331 at 80 mg/kg/day), and G6 (M 10 51–60—10 NRM-331 at 100 mg/kg/day). The experimental groups included a saline vehicle control (CTL), a negative control group receiving 10 mg/kg/day scopolamine (SPA), a positive control group receiving 2 mg/kg/day Aricept (ACT), and treatment groups receiving NRM-331 at varying dosages (40 mg/kg/day, 80 mg/kg/day, and 100 mg/kg/day).

### 4.3. Drug Preparation and Treatments

Scopolamine (SPA) was obtained from Sigma-Aldrich (cat. no. S1875; Burlington, MA, USA) and was prepared in physiological saline at a concentration of 1 mg/10 mL, then injected intraperitoneally at a dose of 1 mg/kg. The SPA was administered 30 min after NRM-331 treatment and 30 min before behavioral tests (Y-maze test, passive avoidance test, and Morris water maze test) for all groups except the control group. Aricept (ACT), a donepezil product from Handok, Republic of Korea, was also prepared in physiological saline at a concentration of 1 mg/10 mL, injected intraperitoneally at a dose of 2 mg/kg, 30 min before the SPA injection, but only for the ACT group.

The NRM-331 (Sigma-Aldrich, cat. No. 247774, Burlington, MA, USA) was used for anti-amnesia treatment and prepared in physiological saline for administration by oral gavage, using a 1 mL syringe equipped with a zonda. NRM-331 was administered 30 min prior to SPA injection for all groups except the CTL group. The treatment duration was set for four weeks, administered daily (seven times a week) from day 1 to day 28, with mice being sacrificed on day 29. Body weights were recorded at the start of acclimatization (day −7), the time of grouping (day 0), and then weekly thereafter (day 1, 7, 14, 21, and 28), along with a final measurement at the time of sacrifice (day 29). Throughout the drug administration and study period, any notable general symptoms or fatalities were recorded.

### 4.4. Behavioral Tests

The Y-maze test was conducted on day 7 following the administration of NRM-331 using the Y-maze test setup (Smart, version 3.0.06, Panlab Harvard Apparatus, Holliston, MA, USA). The experimental apparatus consists of a Y-shaped maze made from transparent acrylic plates (dimensions: W 10 × L 41 × H 25 cm). Each arm of the maze was positioned at a constant angle of 120°. After designating the arms as areas A, B, and C, the experimental animal was carefully placed in one of the areas and allowed to move freely for 8 min. During this time, the animal’s location was recorded by a camera mounted on the ceiling. An entry was counted when the animal’s tail completely crossed into an area, and both the number and the order of entries into each section were measured to evaluate changes in behavioral patterns (spontaneous alternation percentage). A sequence of entries into three different areas was counted as one point (actual alternation, such as ABC, BCA, CAB, etc.). If the animal did not enter consecutively, it was not scored. The percentage of spontaneous alternation was calculated using the following formula:

% Spontaneous alternation = (total number of alternations/(total number of entrances − 2)) × 100.

A passive avoidance test was performed on days 14 and 15 after the administration of NRM-331 using the passive avoidance test setup (cat no. RNV-253C, SG-716B, SG6080D, St. Albans, VT, USA). This setup comprised two compartments divided by a partition with a guillotine door in the middle. One compartment was illuminated, while the other remained dark. The floor of the compartments was gridded and capable of delivering an electric shock. The experiments were conducted at the same time over three consecutive days with 24-h intervals. Initially, on day 13 following NRM-331 administration, animals were allowed to stay in the shaded area for 2 min, after which they were returned to the illuminated area and then immediately taken back to the shaded area for adaptation training. After 24 h, on day 14, SPA was injected 30 min following the administration of NRM-331. Training occurred twice, with 2-min intervals after the SPA injection (acquisition trial). During the first training session, animals were placed in the passive avoidance test chamber for 60 s to adapt to the equipment. At this stage, the guillotine door was opened without illumination, allowing the animal to enter and exit freely. After this 60-s adaptation period, illumination was restored, and after an additional 120 s of free entry and exit, the guillotine door was closed while the animal received a 0.20 mA scrambled shock for 2 s. Finally, 24 h later on day 15 after NRM-331 administration, SPA was again injected 30 min following NRM-331, and the time taken for the animal to move to the shaded area was measured (retention trial).

The Morris water maze test was conducted from day 22 to day 28 following the administration of NRM-331, using the Morris water maze test setup (Smart, version 3.0.06, Panlab Harvard Apparatus, Holliston, MA, USA). In this setup, a platform was submerged in a pool at one of four designated release points within the water tank (diameter: 1 m) and was intended to be located within 60 s. After locating the platform, the animal was allowed to rest on it for about 30 s. If the platform was not found within 60 s, the animal was placed on the platform and allowed to rest for 30 s. Following a trial involving five animals, the next trial commenced, with each trial conducted twice daily. The release points were randomly selected to avoid overlap. The time taken to find the platform was recorded over a period of 7 days (training: 2 days, behavioral experiment: 4 days, probe trial: 1 day). On the final day of the Morris water maze test (day 28), the platform was removed, and a probe trial was carried out for 60 s to measure the number of crosses at the previous platform location. As noted earlier, SPA was injected 30 min following the administration of NRM-331 for the Morris water maze test.

### 4.5. Biochemical Analysis: Serum and Brain

Blood samples were collected and transferred into 5 mL vacuum tubes containing a clot activator (Becton, Dickinson and Company, cat. no. 367955, Franklin Lakes, NJ, USA). The samples were left at room temperature for 15 to 20 min to allow clotting, followed by centrifugation at 3000 rpm for 10 min at 4 °C to separate the serum. The resulting serum was collected in sterilized pre-labeled EP tubes and stored in a −80 °C deep freezer for further molecular analysis. When needed, the serum samples were thawed on ice and used for the quantification of amyloid-β 1-40 (IBL Co. Ltd., cat. no. 27720, Fujioka-Shi, Japan) and amyloid-β 1-42 (IBL Co., Ltd., cat. no. 27721, Fujioka-hi, Japan) using ELISA kits, following the manufacturer’s instructions.

Brain samples were also thawed and homogenized in ice-cold PBS (the ratio of brain weight to PBS volume, PBS molarity, and PBS pH to be specified). After homogenization, the samples were centrifuged at [rpm/min], and the supernatants were collected for acetylcholine (ACh) and acetylcholinesterase (AChE) analysis, which was conducted within 24 h. ACh (MyBiosource, cat. no. MBS733116, San Diego, CA, USA) and AChE (MyBiosource, cat. no. MBS721845, San Diego, CA, USA) levels were determined using ELISA kits, following the manufacturer’s instructions.

### 4.6. Histopathology of the Brain

Prefixed brain samples (in 10% neutral buffered formalin) were prepared for Nissl staining and p-tau antibody analysis to evaluate neuronal plasticity and degeneration histopathologically. The examination focused on the areas of Cornu Ammonis 1 (CA1), Cornu Ammonis 3 (CA3), and the dentate gyrus (DG) in the hippocampus. The prefixed brain was trimmed, and the hippocampal region was processed for paraffin embedding to form paraffin blocks. Tissue blocks were sectioned to a thickness of 4 μm and placed on glass slides. The slides were stained with Nissl stain following the reference protocol and observed using an optical microscope (BX61, Olympus, Tokyo, Japan). Photographs were taken using a digital camera (DP80, Olympus, Tokyo, Japan) [13].

For quantification, the number of Nissl-stained cells in the CA1, CA3, and DG regions of the hippocampus was counted under a high-resolution microscope (at 400× magnification), and Image-Pro software (Media Cybernetics, version 10.0.15; Rockville, MD, USA) was employed for quantitative analysis of the stained cells compared to standard values after measuring positive signal intensity.

For p-tau immunohistochemical staining (IHC), the paraffin sections (4 μm thick) were mounted on SuperFrost glass slides (Thermo Fisher, cat. no. 22037246, Waltham, MA, USA). After deparaffinization with xylene and hydration using ethanol, slides were immersed in 0.01 M citric acid buffer for antigen retrieval and subjected to high temperature and pressure treatment for 10 min in a pressure cooker. Endogenous peroxidase activity was blocked using 3% H_2_O_2_ for 15 min. Following treatment with blocking agents (Vector Laboratories, cat. no. S-1000-20, Newark, CA, USA), anti-p-tau (Ser202, Thr205) antibody (Thermo Fisher, cat. no. MN1020, Waltham, MA, USA) was diluted 1:500 in 1x PBS buffer, incubated for 16 h, and washed three times with 1× PBS buffer. Detection was carried out using the Vectastain Elite ABC kit (Vector Laboratories, cat no. PK-6100, Newark, CA, USA) and 3,3′-Diaminobenzidine, followed by counterstaining with hematoxylin. Observations were made using an optical microscope (BX61, Olympus, Tokyo, Japan), and images were captured using a digital camera (DP80, Olympus, Tokyo, Japan).

For quantitative analysis of the immunohistochemical staining results, the captured images were analyzed for relative ratios based on standard values after measuring positive signal intensity using Image-Pro software (Media Cybernetics, version 10.0.15; Rockville, MD, USA).

### 4.7. Statistical Analysis

Statistical analyses were performed using SPSS (version 12.0K; SPSS Inc., Chicago, IL, USA,). Comparisons of the control (CTL) and study groups (SPA) involved parametric multiple comparison procedures to determine disease causation. The SPA and the APT were compared using a Student’s *t*-test, with Welch’s *t*-test applied when variances were unequal. A one-way ANOVA was utilized to compare the SPA across different administration groups (NRM-331, N40, N80, N100), followed by Duncan’s test for equal variance and Dunnett’s test for non-equal variance based on post-mortem analysis. Statistical significance was determined at *p* < 0.05.

## Figures and Tables

**Figure 1 ijms-26-04613-f001:**
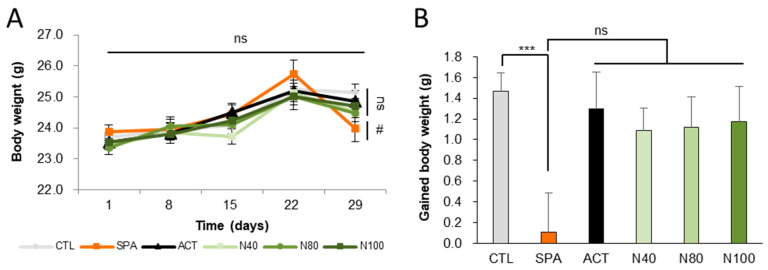
Body weights and gained body weight during the study. (**A**) Weekly body weight. (**B**) Gained body weight. Data were represented by mean ± S.E. The results were statistically analyzed by Welch’s *t*-test and ONE-WAY ANOVA. The SPA group was compared to the CTL group, with significance indicated by *; the remaining groups were compared to the SPA group, with significance indicated by #. *** *p* < 0.001, # *p* < 0.05, ns: not significant. CTL (saline vehicle control); SPA (2 mg/kg/day scopolamine, negative control); ACT (2 mg/kg/day Aricept, positive control); N40 (40 mg/kg/day NRM-331); N80 (80 mg/kg/day NRM-331); N100 (100 mg/kg/day NRM-331).

**Figure 2 ijms-26-04613-f002:**
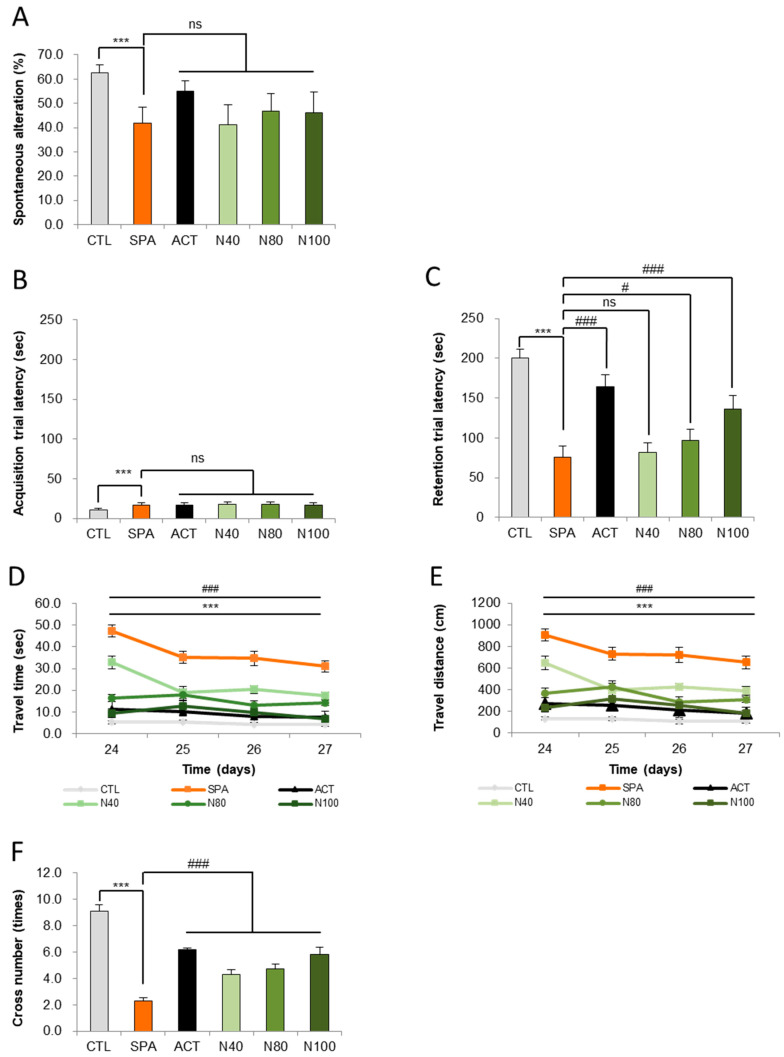
Behavioral tests of the Y maze, passive avoidance, and Morris water maze test. (**A**) Spontaneous alterations in the Y-maze test, (**B**) escape latencies in the passive avoidance test, (**C**) escape latencies in the Morris water maze test, (**D**) swim time in the Morris water maze test, (**E**) swim distances in the Morris water maze test, (**F**) number of platform crossing in the Morris water maze test. Data were represented by mean ± S.E. The results were statistically analyzed by ONE-WAY and TWO-WAY ANOVA. SPA group was compared to the CTL group, with significance indicated by *; the remaining groups were compared to SPA, with significance indicated by #. *** or ### *p* < 0.001, # *p* < 0.05, ns: not significant. CTL (saline vehicle control); SPA (2 mg/kg/day scopolamine, negative control); ACT (2 mg/kg/day Aricept, positive control); N40 (40 mg/kg/day NRM-331); N80 (80 mg/kg/day NRM-331); N100 (100 mg/kg/day NRM-331).

**Figure 3 ijms-26-04613-f003:**
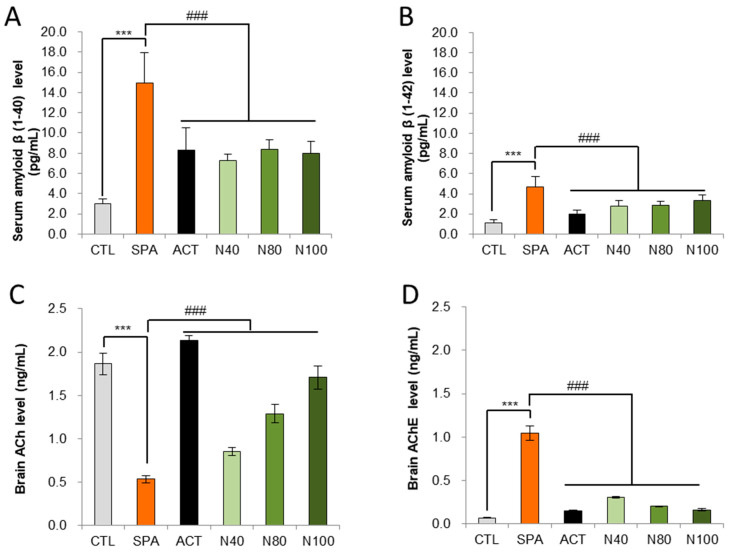
Biochemical analysis in serum for Aβ 1-40 and Aβ 1-42 levels and in brain tissue for ACh and AChE levels by ELISA. (**A**) Serum amyloid-β 1-40 level, (**B**) Serum amyloid- β 1-42 level, (**C**) Brain tissue ACh level, and (**D**) Brain tissue AChE level. The results were statistically analyzed by ONE-WAY ANOVA. The SPA group was compared to the CTL group, with significance indicated by *; the remaining groups were compared to SPA, with significance indicated by #. *** or ### *p* < 0.001, ns: not significant. CTL (saline vehicle control); SPA (2 mg/kg/day scopolamine, negative control); ACT (2 mg/kg/day Aricept, positive control); N40 (40 mg/kg/day NRM-331); N80 (80 mg/kg/day NRM-331); N100 (100 mg/kg/day NRM-331); Aβ: Amyloid beta; ACh: Acetylcholine; AChE: Acetylcholine esterase.

**Figure 4 ijms-26-04613-f004:**
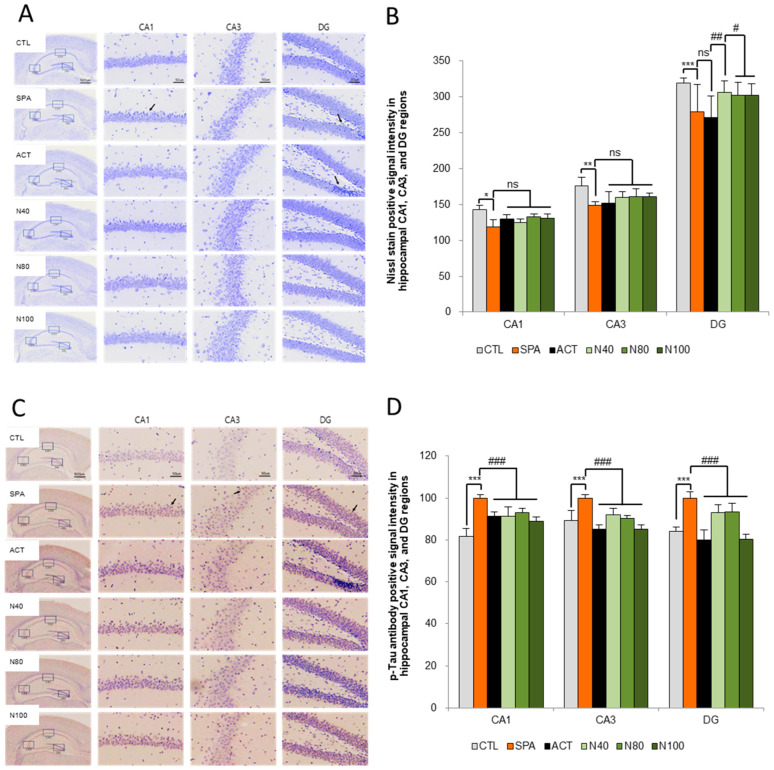
Histopathology of the brain for the Nissl stain and tau-accumulation by ELISA. (**A**) Representative images of Nissl staining neuronal cells in the hippocampus, (**B**) Quantification of Nissl stain, (**C**) Representative images of immunohistochemical staining for p-tau, and (**D**) Quantification of p-tau. The results were statistically analyzed by ONE-WAY and TWO-WAY ANOVA. SPA group was compared with the CTL group, and the remaining groups were compared with the SPA group. *** or ^###^: *p* < 0.001, ** or ^##^: *p* < 0.01, * or ^#^: *p* < 0.05, ns: not significant, Arrow: shrunken neuron or degenerated neuronal cell, or p-tau signal intensity. Scale bars: 500 μm or 50 μm. CTL (saline vehicle control); SPA (2 mg/kg/day scopolamine, negative control); ACT (2 mg/kg/day Aricept, positive control); N40 (40 mg/kg/day NRM-331); N80 (80 mg/kg/day NRM-331); N100 (100 mg/kg/day NRM-331).

## Data Availability

All data is contained within the article with Appendix A.

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
