# Peer review of "Sobrerol Improves Memory Impairment in the Scopolamine-Induced Amnesia Mouse Model"

_ijms, 2025, doi:10.3390/ijms26104613_

Round 1

Reviewer 1 Report

Comments and Suggestions for Authors

This manuscript addresses the neuroprotective effect of sobrerol (NRM-331) in a scopolamine-induced mouse model of amnesia. The study employs standard behavioral, biochemical, and histological methods to examine memory impairment and neurodegeneration, making the topic timely and relevant for AD therapeutics. However, there are numerous grammatical issues, inconsistent formatting, repetition, and unclear phrasing throughout the manuscript, which severely hamper readability and scientific rigor. Substantive scientific claims also lack mechanistic depth and clarity in several instances.

  1. The abstract mixes past and present tense inconsistently and contains inaccuracies. Revise abstract for tense consistency, grammar, and clarity. For example:
  • Page 1, Line 20: “passive avidity test, and Morris water test” → should be “passive avoidance test, and Morris water maze test.”
  • Line 23: “passive avoidance teste” → should be “test.”
  • Line 24: “improvements were notes” → should be “noted.”
  • Line 25: “NRM-331groups” → missing space.
  1. Frequent grammar issues, improper phrasing, and awkward sentence structures. A thorough language editing by a native English-speaking editor or language editing service is recommended before resubmission. For example:
  • Page 2, Line 98: "passive avidity test, and Morris water test" → It should be "passive avoidance test, and Morris water Maze test"
  • Page 9, Line 262-263: "Our findings form the Y-maze test" →should be “from the Y-maze test.”
  1. The author states that "each group was kept in separate cages" (Page 12, Line 379), and each group contained 10 mice. This implies that 10 mice may have been housed per cage, which exceeds standard housing density limits for adult mice unless special large housing was used. This appears to conflict with the earlier statement that "five animals were housed per polycarbonate cage" (Page 11, Line 355). The authors should clarify the actual housing conditions and ensure they complied with institutional IACUC guidelines. If standard caging was used for 10 mice, this would constitute an animal welfare and regulatory violation.
  2. In Figure 2F, the term“cross number” is confusing and not standard terminology. It would be clearer and more appropriate to use “number of platform crossings” to describe the measure in the Morris water maze probe trial.
  3. The study lacks a group treated with NRM-331 alone (i.e., without scopolamine administration). All treatment groups include scopolamine-induced amnesia, which makes it impossible to determine whether NRM-331 has inherent cognitive-enhancing effects or if its benefits are strictly limited to a pathological context. The absence of a baseline NRM-331 control group prevents the differentiation between preventive and therapeutic actions of the compound.
  4. The results of the behavioral experiments are not coherent across different memory domains. The Y-maze test showed no improvement, the passive avoidance test revealed only dose-dependent partial effects, while the Morris water maze demonstrated robust changes. These inconsistencies are not addressed in the discussion and weaken the behavioral conclusions.
  5. There is no attempt to correlate changes in ACh, AChE, or Aβ levels with behavioral improvements in discussion section. Such correlations are critical to substantiate mechanistic interpretations and link molecular changes to functional outcomes.
  6. ACh and AChE levels were measured in whole brain homogenates. Since memory functions are mediated primarily through the hippocampus, region-specific data would provide more meaningful insights.
  7. There are multiple instances of incorrect or ambiguous statistical notations, such as “(P < 0.05 or P < 0.01)” (Page 7, Line 220) and “P > 0.05 or P > 0.01” (Page 4, Line 145), which are unclear and contradict standard interpretation. The authors should revise all p-value reporting to ensure clarity, consistency, and accuracy. Each result should be reported with a specific significance threshold or, ideally, the exact p-value. Accurate statistical representation is essential for reliable data interpretation.
  8. Although sobrerol is stated to have antioxidant properties, no oxidative stress or inflammatory markers (e.g., ROS, IL-6, TNF-α, BDNF, CREB) are assessed. These should have been measured or at least discussed as key mechanisms underlying cognitive protection.
  9. The reduction in p-tau levels is a major finding, but there is no discussion of the underlying signaling pathways that may be involved (e.g., inhibition of kinases like GSK-3β or CDK5). This weakens the impact of this result.
  10. ACh and AChE levels were measured in whole brain homogenates. Since memory functions are mediated primarily through the hippocampus, region-specific data would provide more meaningful insights.
  11. There are inconsistencies in group naming throughout the manuscript. For example, “Aricept” is referred to as “APT” on Page 1, Line 18, but as “ACT” elsewhere. Similarly, “SPL” appears in place of “SPA” (Scopolamine) on Page 4, Lines 137–145. The use of group abbreviations should be standardized across the text, figures, and legends for clarity and consistency.
Comments on the Quality of English Language

The manuscript requires significant language editing to improve clarity, grammar, and readability. Throughout the text, there are numerous instances of awkward phrasing, tense inconsistency, typographical errors, and unclear sentence structures.

Author Response

Comments and Suggestions for Authors

This manuscript addresses the neuroprotective effect of sobrerol (NRM-331) in a scopolamine-induced mouse model of amnesia. The study employs standard behavioral, biochemical, and histological methods to examine memory impairment and neurodegeneration, making the topic timely and relevant for AD therapeutics. However, there are numerous grammatical issues, inconsistent formatting, repetition, and unclear phrasing throughout the manuscript, which severely hamper readability and scientific rigor. Substantive scientific claims also lack mechanistic depth and clarity in several instances.

Comment

  1. The abstract mixes past and present tense inconsistently and contains inaccuracies. Revise abstract for tense consistency, grammar, and clarity. For example:
  • Page 1, Line 20: “passive avidity test, and Morris water test” → should be “passive avoidance test, and Morris water maze test.”
  • Line 23: “passive avoidance teste” → should be “test.”
  • Line 24: “improvements were notes” → should be “noted.”
  • Line 25: “NRM-331groups” → missing space.

Reply 1: We corrected these grammatical mistakes in revised version, thank you.

Comment:

  1. Frequent grammar issues, improper phrasing, and awkward sentence structures. A thorough language editing by a native English-speaking editor or language editing service is recommended before resubmission. For example:
  • Page 2, Line 98: "passive avidity test, and Morris water test" → It should be "passive avoidance test, and Morris water Maze test"
  • Page 9, Line 262-263: "Our findings form the Y-maze test" →should be “from the Y-maze test.”

Reply 2: We corrected these grammatical mistakes in revised version, thank you.

Comment:

  1. The author states that "each group was kept in separate cages" (Page 12, Line 379), and each group contained 10 mice. This implies that 10 mice may have been housed per cage, which exceeds standard housing density limits for adult mice unless special large housing was used. This appears to conflict with the earlier statement that "five animals were housed per polycarbonate cage" (Page 11, Line 355). The authors should clarify the actual housing conditions and ensure they complied with institutional IACUC guidelines. If standard caging was used for 10 mice, this would constitute an animal welfare and regulatory violation.

Reply 3: As per page 11, line 355, 5 animals were housed in one cage during acclimatization period. After grouping, similarly, five animals were housed per cage and cages were labeled with unique color codes’. We clarify in revised version, thank you. 

Comment:

  1. In Figure 2F, the term“cross number” is confusing and not standard terminology. It would be clearer and more appropriate to use “number of platform crossings” to describe the measure in the Morris water maze probe trial.

Reply 4: We incorporated your suggestion in revised version, thank you.

Comment:

  1. The study lacks a group treated with NRM-331 alone (i.e., without scopolamine administration). All treatment groups include scopolamine-induced amnesia, which makes it impossible to determine whether NRM-331 has inherent cognitive-enhancing effects or if its benefits are strictly limited to a pathological context. The absence of a baseline NRM-331 control group prevents the differentiation between preventive and therapeutic actions of the compound.

Reply 5: Thank you for this valuable comment. You raise an important point regarding the absence of a NRM-331 only group. As this is a preliminary study focused on optimizing the effective dose range (N-40, N-80, and N-100), our primary aim was to evaluate whether NRM-331 can reverse memory impairment in a scopolamine-induced amnesia model. We did not intend to assess its potential cognitive-enhancing effects in a non-pathological (healthy) context at this stage. Including an additional set of groups receiving NRM-331 alone would have significantly increased the number of animals required. In alignment with ethical research practices and the 3Rs principle (Reduce, Refine, Replace), we prioritized minimizing animal use while still allowing for a robust assessment of dose-response in the disease model. That said, we fully acknowledge the importance of assessing the inherent cognitive effects of NRM-331 in non-diseased models. This is planned for future studies where both preventive and therapeutic effects will be more comprehensively explored.

Comment:

  1. The results of the behavioral experiments are not coherent across different memory domains. The Y-maze test showed no improvement, the passive avoidance test revealed only dose-dependent partial effects, while the Morris water maze demonstrated robust changes. These inconsistencies are not addressed in the discussion and weaken the behavioral conclusions.

Reply 6: Yes, in the behavioral experiments Y-maze test showed no improvement, the passive avoidance test revealed only dose-dependent partial effects, while the Morris water maze demonstrated robust changes. Though inconsistencies, but there the improvement tendency can were seen, which discussed in revised version.

Comment:

  1. There is no attempt to correlate changes in ACh, AChE, or Aβ levels with behavioral improvements in discussion section. Such correlations are critical to substantiate mechanistic interpretations and link molecular changes to functional outcomes.

Reply 7: Yes, he levels of ACh, AChE, and Aβ are closely connected to performance in behavioral tests, we incorporated in discussion section. “ACh is crucial for synaptic plasticity and memory formation. Higher ACh levels are generally associated with better performance in behavioral tests such as the Morris water maze, Y-maze, or novel object recognition, which assess spatial learning and memory. AChE breaks down ACh in the synaptic cleft. Elevated AChE activity can reduce ACh availability, potentially impairing cognitive function. Therefore, reduced AChE activity often correlates with improved performance in memory-related tasks, supporting the efficacy of AChE inhibitors in enhancing cognition. On the other hand Aβ, accumulates abnormally in AD and contributes to synaptic dysfunction and neurodegeneration. Higher Aβ levels typically correlate with poorer behavioral performance. Hence, a decrease in Aβ burden, accompanied by improvements in learning and memory tasks, suggests that interventions may be neuroprotective or disease-modifying”.

Comment:

  1. ACh and AChE levels were measured in whole brain homogenates. Since memory functions are mediated primarily through the hippocampus, region-specific data would provide more meaningful insights.

Reply: 8: Thank you for your insightful comment. You are absolutely right that memory functions are primarily mediated through the hippocampus, and the cortex also plays a significant role in memory-related cholinergic activity. However, AChE is widely distributed throughout the brain and is produced by both cholinergic neurons and certain non-cholinergic neurons as well as glial cells. Since AChE plays a critical role in breaking down ACh in the synaptic cleft, we chose to measure ACh and AChE levels in whole brain homogenates to provide an overall assessment of cholinergic activity. We agree that region-specific analysis, particularly in the hippocampus and cortex, would yield more targeted insights and will consider this approach in future studies.

Comment:

  1. There are multiple instances of incorrect or ambiguous statistical notations, such as “(P < 0.05 or P < 0.01)” (Page 7, Line 220) and “P > 0.05 or P > 0.01” (Page 4, Line 145), which are unclear and contradict standard interpretation. The authors should revise all p-value reporting to ensure clarity, consistency, and accuracy. Each result should be reported with a specific significance threshold or, ideally, the exact p-value. Accurate statistical representation is essential for reliable data interpretation.

Reply 9: We corrected all p-value in revised version, thank you.

Comment:

  1. Although sobrerol is stated to have antioxidant properties, no oxidative stress or inflammatory markers (e.g., ROS, IL-6, TNF-α, BDNF, CREB) are assessed. These should have been measured or at least discussed as key mechanisms underlying cognitive protection.

Reply 10: Thank you for your valuable comment. You are correct that “sobrerol consist antioxidant properties, and oxidative stress and inflammation are key mechanisms involved in cognitive decline. In the current study, our primary objective was to investigate the effects of sobrerol on cholinergic markers (ACh, AChE), as well as neuropathological markers such as p-tau and Aβ, in relation to behavioral performance. While we did not assess oxidative stress or inflammatory markers (e.g., ROS, IL-6, TNF-α, BDNF, CREB), would have provided a more comprehensive understanding of the mechanisms underlying sobrerol’s cognitive protective effects. While we did not assess oxidative stress or inflammatory markers, which is the limitation of this study”.

Comment:

  1. The reduction in p-tau levels is a major finding, but there is no discussion of the underlying signaling pathways that may be involved (e.g., inhibition of kinases like GSK-3β or CDK5). This weakens the impact of this result.

Reply 11: Thank you for your appreciation, yes, p-tau is a major finding, we described the GSK-3B and CDK5 in discussion part with relevant references.

Comment:

  1. ACh and AChE levels were measured in whole brain homogenates. Since memory functions are mediated primarily through the hippocampus, region-specific data would provide more meaningful insights.

Reply 13: Same comment as comment 8. Thank you for your insightful comment. You are absolutely right that memory functions are primarily mediated through the hippocampus, and the cortex also plays a significant role in memory-related cholinergic activity. However, AChE is widely distributed throughout the brain and is produced by both cholinergic neurons and certain non-cholinergic neurons as well as glial cells. Since AChE plays a critical role in breaking down ACh in the synaptic cleft, we chose to measure ACh and AChE levels in whole brain homogenates to provide an overall assessment of cholinergic activity. We agree that region-specific analysis, particularly in the hippocampus and cortex, would yield more targeted insights and will consider this approach in future studies.

Comment:

  1. There are inconsistencies in group naming throughout the manuscript. For example, “Aricept” is referred to as “APT” on Page 1, Line 18, but as “ACT” elsewhere. Similarly, “SPL” appears in place of “SPA” (Scopolamine) on Page 4, Lines 137–145. The use of group abbreviations should be standardized across the text, figures, and legends for clarity and consistency.

Reply 13: We incorporated the abbreviations throughout the manuscript, thank you.

Reviewer 2 Report

Comments and Suggestions for Authors

A peer-reviewed study evaluated the potential effect of sobrerol (NRM-331) in alleviating memory impairment and neuronal dysfunction in a mouse model of scopolamine(SPA)-induced amnesia. Sobrerol is an approved mucolytic agent with additional antioxidant and immunomodulatory effects [https://doi.org/10.3390/children10071210]. In the described experiment, the authors performed several behavioral tests (Y-maze test, passive avoidance test, and Morris water test), biochemical analysis (in serum Aß 1-40 and Aß 1-42, and in brain ACh and AChE), and brain histopathological analysis (Nissl staining and tau IHC) in rodents. They obtained results showing that sobrerol improves memory and cognitive impairment in a mouse model of amnesia. Decreased serum Aβ levels, increased brain ACh levels, and decreased hippocampal p-tau accumulation confirm the improvement of disease symptoms, as these factors reflect increased neuronal plasticity. In conclusion, these data show the therapeutic potential of sobrerol in neurodegenerative diseases such as Alzheimer's disease by targeting key factors that contribute to memory impairment and neuronal degeneration.

I have only a few minor comments to improve the text. They are as follows:

- In the introduction, the authors presented several compounds and phytochemicals with therapeutic potential in Alzheimer's disease. However, there is no brief information on the chemical structure, molecular mechanism, and pharmacological properties of sobrerol (it is not a plant compound). In line with the premise that aging is the predominant risk factor for AD, many studies have emphasized that increased levels of oxidative stress, mitochondrial dysfunction, neuroinflammation, and metabolic changes may be critical in AD. Therefore, information in this area should be specifically described for sobrerol.

Lines 73 and 78 - an error has crept in in writing the date "investigated in the 1050s" and the name of the substance "Couarin" (? coumarin); please write the names of the compounds in lower case, e.g., chlorogenic acid, coumarin, ginsenosides, donepezil, etc.

Lines 76 and 79 - plant species names should be given in full with a citation, i.e, the name of the author of the name, e.g., Callicarpa dichotoma (Lour.) K.Koch, or Bacopa monnieri (L.) Wettst.; the first two parts of the species name are in italics;

Lines 138 and 264 - instead of SPA, a different notation "SPL" or "SAP" has been used.

Author Response

Comments and Suggestions for Authors

A peer-reviewed study evaluated the potential effect of sobrerol (NRM-331) in alleviating memory impairment and neuronal dysfunction in a mouse model of scopolamine (SPA)-induced amnesia. Sobrerol is an approved mucolytic agent with additional antioxidant and immunomodulatory effects [https://doi.org/10.3390/children10071210]. In the described experiment, the authors performed several behavioral tests (Y-maze test, passive avoidance test, and Morris water test), biochemical analysis (in serum Aß 1-40 and Aß 1-42, and in brain ACh and AChE), and brain histopathological analysis (Nissl staining and tau IHC) in rodents. They obtained results showing that sobrerol improves memory and cognitive impairment in a mouse model of amnesia. Decreased serum Aβ levels, increased brain ACh levels, and decreased hippocampal p-tau accumulation confirm the improvement of disease symptoms, as these factors reflect increased neuronal plasticity. In conclusion, these data show the therapeutic potential of sobrerol in neurodegenerative diseases such as Alzheimer's disease by targeting key factors that contribute to memory impairment and neuronal degeneration.

I have only a few minor comments to improve the text. They are as follows:

- In the introduction, the authors presented several compounds and phytochemicals with therapeutic potential in Alzheimer's disease. However, there is no brief information on the chemical structure, molecular mechanism, and pharmacological properties of sobrerol (it is not a plant compound). In line with the premise that aging is the predominant risk factor for AD, many studies have emphasized that increased levels of oxidative stress, mitochondrial dysfunction, neuroinflammation, and metabolic changes may be critical in AD. Therefore, information in this area should be specifically described for sobrerol.

Reply: Thank you for your suggestion, we incorporated the chemical structure, and molecular mechanism and pharmacological properties in revised version, thank you.

Lines 73 and 78 - an error has crept in in writing the date "investigated in the 1050s" and the name of the substance "Couarin" (? coumarin); please write the names of the compounds in lower case, e.g., chlorogenic acid, coumarin, ginsenosides, donepezil, etc.

Reply: We incorporate the mistake and suggestion, thank you.

Lines 76 and 79 - plant species names should be given in full with a citation, i.e, the name of the author of the name, e.g., Callicarpa dichotoma (Lour.) K.Koch, or Bacopa monnieri (L.) Wettst.; the first two parts of the species name are in italics;

Reply: We incorporated your suggestion, thank you.

Lines 138 and 264 - instead of SPA, a different notation "SPL" or "SAP" has been used.

Reply: We corrected the abbreviation throughout the manuscript, thank you.

Round 2

Reviewer 1 Report

Comments and Suggestions for Authors

Thank you for your careful revisions. You've addressed the scientific and editorial concerns thoroughly, and the manuscript is now much clearer and more coherent. While some limitations remain, your explanations and future directions are appropriate. I appreciate your efforts and wish you success with this line of research.